# Low-Risk Women with Suspicious Microcalcifications in Mammography—Can an Additional Breast MRI Reduce the Biopsy Rate?

**DOI:** 10.3390/diagnostics13061197

**Published:** 2023-03-22

**Authors:** Patrik Pöschke, Evelyn Wenkel, Carolin C. Hack, Matthias W. Beckmann, Michael Uder, Sabine Ohlmeyer

**Affiliations:** 1Department of Gynecology and Obstetrics, University Hospital Erlangen, Universitätsstraße 21-23, 91054 Erlangen, Germany; 2Radiologie München, Burgstraße 7, 80331 München, Germany; 3Medizinische Fakultät, Friedrich-Alexander-Universität Erlangen-Nürnberg (FAU), 91054 Erlangen, Germany; 4Department of Radiology, University Hospital Erlangen, Maximiliansplatz 3, 91054 Erlangen, Germany

**Keywords:** breast magnetic resonance imaging (MRI), breast cancer, screening, ductal carcinoma in situ (DCIS), mammography

## Abstract

Background: In the German Mammography Screening Program, 62% of ductal carcinoma in situ (DCIS) and 38% of invasive breast cancers are associated with microcalcifications (MCs). Vacuum-assisted stereotactic breast biopsies are necessary to distinguish precancerous lesions from benign calcifications because mammographic discrimination is not possible. The aim of this study was to investigate if breast magnetic resonance imaging (MRM) could assist the evaluation of MCs and thus help reduce biopsy rates. Methods: In this IRB-approved study, 58 women (mean age 58 +/− 24 years) with 59 suspicious MC clusters in the MG were eligible for this prospective single-center trial. Additional breast magnetic resonance imaging (MRI) was conducted before biopsy. Results: The breast MRI showed a sensitivity of 86%, a specificity of 84%, a positive predictive value (PPV) of 75% and a negative predictive value (NPV) of 91% for the differentiation between benign and malignant in these 59 MCs found with MG. Breast MRI in addition to MG could increase the PPV from 36% to 75% compared to MG alone. The MRI examination led to nine additional suspicious classified lesions in the study cohort. A total of 55% (5/9) of them turned out to be malignant. A total of 32 of 59 (54 %) women with suspicious MCs and benign histology were classified as non-suspicious by MRI. Conclusion: An additionally performed breast MRI could have increased the diagnostic reliability in the assessment of MCs. Further, in our small cohort, a considerable number of malignant lesions without mammographically visible MCs were revealed.

## 1. Introduction

Breast cancer is the most common cancer in women, with a lifetime risk of 13%, which means that one in eight women will develop breast cancer [1]. The early detection of breast cancer is the mainstay of mammography screening programs, which aim to reduce the number of advanced carcinomas and breast cancer mortality. Currently, women in Germany between the ages of 50 and 69 are invited to a population-based mammography screening program every two years. A meta-analysis in the US that examined breast cancer screening over ten years showed that the rate of breast cancer deaths per 10,000 women could be reduced by eight women in patients aged 50–59, and by 21 women between the ages of 60 and 69 [2].

For the overall survival of the patients, it would be optimal to detect the disease before invasiveness, for example, in the precancerous stage as a ductal in situ carcinoma (DCIS). A total of 31% of the findings in mammography screenings contain suspected MCs. In the histopathological evaluation of these MCs, only 14% turn out to be malignant [3,4]. A total of 79% of DCIS show suspicious MCs using mammography (MG), but the specificity of MG regarding MCs is low, ranging between 24% and 59% in the literature [5,6,7]. Therefore, the current reference standard is histopathological evaluation by biopsy. This results in overdiagnosis and therapy, which is a major source of criticism [8,9].

A method with high sensitivity and specificity without burdening women with unnecessary biopsies would be desirable [10]. Breast MRI evaluations, especially in high-risk patients, have a high sensitivity not only for invasive carcinoma from 90% to 99% and a specificity from 72% to 88% but also for calcified and non-calcified DCIS with a sensitivity from 59% to 92% and a specificity from 62% to 97% [11,12,13,14,15,16]. In addition, it was shown that occult breast cancers that were not detectable in MG or ultrasound could be found using MRI [11]. Therefore, MRI showed a survival benefit [17,18,19], especially in women with an increased risk of breast cancer and the European Society of Breast Imaging (EUSOBI) recommends offering breast MRI to women with extremely dense breasts [20].

The aim of this study was to investigate if additional breast MRI in women with suspicious MCs via MG can reduce the biopsy rate. Furthermore, it should be evaluated whether additional breast MRI in these women leads to underdiagnoses by missing an invasive carcinoma or possibly to overdiagnosis and therapy.

## 2. Materials and Methods

### 2.1. Trial Design and Population

This trial was a prospective single-center study, approved by the institutional ethics committee. The inclusion criteria were suspicious MCs on mammography, BI-RADS^®^ (Breast Imaging Reporting & Data System; American College of Radiology; Reston, VA, USA) category 4 [21] and women between 18 and 80 years of age. The exclusion criteria were mass findings, a palpable tumor in clinical examinations, breast pain, history of breast cancer and BI-RADS^®^ category three and five microcalcifications. Histology was obtained by a stereotactic-guided vacuum-assisted breast biopsy (VABB) or surgery. From October 2014 to July 2016, a histological evaluation of suspicious MCs was indicated in 219 patients. A total of 161 women with suspicious MCs could not take part in the study mainly due to time constraints of the MRI scan time (*n* = 131). A total of 23 women did not want to participate in the study and four had contraindications for the MRI examination. Three of the patients who underwent an MRI examination decided against histological verification of the abnormal MCs and thus were excluded from the final evaluation.

### 2.2. Mammography

MG was performed as digital full-field MG in standard mediolateral oblique and craniocaudal projections [22]. Detection and classification of MCs as BI-RADS^®^ category four lesions were performed by radiologists experienced in breast imaging.

### 2.3. Breast Magnetic Resonance Imaging

Breast MRI was performed with a 3T scanner (Verio, Siemens Healthcare, Erlangen, Germany). A dedicated bilateral 16-channel breast coil was used, with the imaging protocol consisting of the following sequences: a transversal T2 weighted (w) STIR (short-tau inversion recovery) sequence (TR/TE/TI, 3570/70/230 ms, flip angel 80°, voxel size 0.8 × 0.8 × 4.0 mm^3^) followed by a dynamic T1w sequence and a DWI (diffusion weighted imaging) sequence. The parameters for the dynamic VIBE (volume interpolated breath hold examination) Dixon sequence were TR/TE 5.97/2.46 ms, flip angle 10°, voxel size 0.8 × 0.8 × 1.5 mm^3^ with a dynamic recording time of 60 s. Pre- and five post-KM sequences were used for the analysis of the dynamic sequences. As an intravenous contrast agent, Gadobutrol (Gadovist^®^, Bayer Schering Pharma AG, Berlin, Germany) was administered with an injection rate of 2 mL/s weight-adapted at 0.1 mmol/kg body weight. The diffusion-weighted images were conducted as a SPAIR (spectral attenuated inversion recovery) fat suppression sequence (TR/TE, 3800/66 ms, voxel size 1.7 × 1.7 × 5.0 mm^3^). The total duration of the examination was approximately 13 min.

### 2.4. Breast MRI Image Evaluation

Breast MRI datasets were prospectively collected and evaluated by a radiologist with more than 10 years of clinical experience in evaluating breast MRI in the knowledge of the MG results and a second radiologist as part of the clinical routine. Image evaluation was performed as in clinical routine by full protocol, with additional MIP images. Suspicious findings in both modalities were assessed by an experienced breast radiologist (E.W.), associated with the lesions’ location and correlated to histologic results. This evaluation was carried out prospectively prior to the biopsy. The findings detected in the breast MRI were categorized according to the BI-RADS^®^ assessment criteria [23]. Movement artifacts and background parenchymal enhancement (BPE) were recorded. The findings in the MG and in the breast MRI were dichotomized into groups, with all lesions corresponding to category four according to the BI-RADS^®^ classified as suspicious and those in the BI-RADS^®^ category two as benign. All breasts without findings (BI-RADS^®^ category one) were subsumed as benign findings. The BI-RADS^®^ category three was avoided to enable a study evaluation. Results of the final histology were used as the reference standard: true positive (TP) corresponds to abnormal imaging and a malignant histological finding, and true negative (TN) corresponds to normal imaging and a benign histological finding. False negative (FN) means inconspicuous imaging in malignant histology, and false positive (FP) describes suspicious imaging in benign histology.

### 2.5. Statistical Analysis

A two-sided independent t-test and a chi-square test were carried out to differentiate the mean values of the descriptive and image-morphological characteristics for patients with benign or malignant findings. A significance level of 5% was used (*p*-value < 0.05). The Mann–Whitney U-test was performed as a non-parametric test. To determine the additional diagnostic value of breast MRI, the sensitivity, specificity, positive predictive value (PPV), negative predictive value (NPV) and accuracy (ACC) of the combined examination of breast MRI and MG were evaluated. The statistical software IBM SPSS Statistics Version 23.0 (IBM Corp., Armonk, NY, USA) was used for the statistical analysis.

## 3. Results

### 3.1. Breast MRI and MG Examinations

A total of 116 breasts of 58 women were evaluated with breast MRI and MG. A total of 59 were suspicious (BI-RADS^®^ category IV) in the MG and 21 turned out to be histologically malignant. Breast MRI evaluation resulted in 33 suspicious findings (Figure 1).

### 3.2. Characteristics of Trial Population

A total of 58 women were eligible for this prospective study. The mean age was 56 years ranging from 45 to 80 years, and the standard deviation (SD) 7.46 years. All women presented with suspicious MCs and one presented with two clusters. The statistical analysis between the groups of women with benign and malignant pathological findings showed no significant differences in terms of age, menopausal status, average BMI (body mass index) and family history. A significant deviation was only found in terms of breast density (Table 1).

### 3.3. Histology

In total, 68 breast findings from 58 patients were analyzed histologically. A total of 59 of these findings in the 58 women contained suspicious MCs in the MG. Out of 68 histological findings, 42 were benign and 26 were malignant. One woman had bilateral suspicious MCs. Nine additional findings were without MCs, four unilateral findings and five contralateral findings only detected on the MRI are described separately (see histology breast MRI section and Table 2).

### 3.4. Histology of Lesions with Suspicious MCs in MG

Focusing on the suspicious MCs in the MG (*n* = 59), the histopathological results showed 38/59 (64%) benign findings and 21/59 (36%) malignant findings.

The 38 benign findings consisted of fibrocystic mastopathy in ten (26%), common ductal hyperplasia (UDH) in seven (18%), sclerosing adenosis in six (14%), fibroadenoma in six (14%), flat epithelial atypia (FEA) in five (13%), papilloma in two (5%) and atypical ductal hyperplasia (ADH) in one case (2%). In addition, a grade three lobular neoplasia (LN) associated with a papilloma (2%) was found. Six of these lesions were categorized as B3 lesions after biopsy, and the final histology resulted in four FEA, one LN3 and one UDH.

A total of 21 malignancies were diagnosed in 20/58 (34%) of the women, with one woman with bilateral findings:-A total of 14/21 (67%) of the malignant lesions represented a DCIS: high-grade in 7/14 cases (50%), intermediate-grade in one case (7%) and low-grade in six cases (43%).-A total of 7/21 (33%) malignant findings showed an invasive carcinoma. Two exclusively invasive carcinomas (no special type (NST), one G1 and one G2). A total of 5/7 of the invasive carcinomas were associated with DCIS (one NST G1 with low-grade DCIS, two NST G2 each with an intermediate-grade and high-grade DCIS, one NST G3 with a high-grade DCIS and one invasive tubular carcinoma one associated with a high-grade DCIS).

In 35 cases, no suspicious contrast enhancement in the MRI was found at the site of the MCs. A total of 32 of these cases turned out to be benign and in three cases a DCIS was detected (see Table 3 and Figure 2). In Figure 2, the 53-year-old patient number three of the following Table 3 is shown.

### 3.5. Histology of Lesions Additionally Detected in Breast MRI

Nine additional findings without MCs were detected in the breast MRI with consecutive histological evaluation. A total of 5/9 of the additional findings resulted in a malignant and 4/9 a benign finding.

Of these nine additional findings, five were on the contralateral side to the MCs (three malignant, two benign) and four (two malignant, two benign) of these were ipsilateral to the MCs, but at a different location. In the following Table 2, these nine additional findings are listed, together with the resulting histology, the MG finding, the breast MRI finding and the density of the mammary gland tissue. The MG and breast MR images of patient number five in Table 2 are shown in Figure 3. The histology of that patient shows on the right breast a 4.9 cm low-grade DCIS, the left breast shows a 7 mm invasive ductal carcinoma (G2 NST and a 5 cm intermediate-grade DCIS.

### 3.6. Comparison of the Diagnostic Accuracy of MG and MG Plus Breast MRI in Evaluation of MCs

The diagnostic performance of the MG alone in terms of MC evaluation was as follows: false positive in 38 cases, true positive in 21 cases. This resulted in a positive predictive value (PPV) of 36%.

The diagnostic accuracy of breast MRI evaluation of these 59 MCs was as follows: false positive in 6 cases, true positive in 18 cases, false negative in 3 cases and true negative in 32 cases. This resulted in a sensitivity of 86%, a specificity of 84%, a positive predictive value (PPV) of 75% and a negative predictive value (NPV) of 91% for differentiation between benign and malignant in these 59 MCs.

Breast MRI in addition to MG could increase the PPV from 36% to 75% by 39% compared to MG alone.

### 3.7. Results of the Additionally Performed Breast MRI

As a consequence of the additional performed breast MRI, nine additional biopsies were conducted. This resulted in four further benign and five malignant findings (see Table 2). Four of these findings were on the contralateral side of the MCs and five were unilaterally in a different position to the MCs. The diagnostic accuracy of breast MRI was as follows based on histopathological results (*n* = 68): false positive in 10 cases, true positive in 23 cases, false negative in 3 cases and true negative in 32 cases. This results in a sensitivity of 88%, a specificity of 76%, an NPV of 91%, a PPV of 70% and an accuracy of 81%.

## 4. Discussion

In this single-center trial with 58 patients presenting with 59 suspicious MCs in the MG, the additional performed breast MRI was able to improve the diagnostic accuracy for the differentiation of the MCs. The PPV increased from 36 % to 75 % by performing an additional breast MRI in these patients. Breast MRI led to nine additional biopsies, resulting in five accidentally diagnosed breast cancers in our study.

Bennani-Baiti et al. found an additional diagnostic benefit from breast MRI as well, especially for patients with mammographically detected MCs (BI-RADS^®^ category four). In their meta-analysis of 20 studies, they described 1843 findings with suspected MCs, of which 748 (40.6%) were malignant. Pooled sensitivity and specificity were 87% and 81% for all lesions [3]. This corresponds to our results where the additional breast MRI resulted in a sensitivity of 86%, a specificity of 84%, a positive predictive value (PPV) of 75% and a negative predictive value (NPV) of 91% in lesions presenting with MCs. In other comparable studies on breast MRI and MCs in mammography, collectives of patients with an increased risk of breast cancer, for example in dense FGT or family high-risk situations, were examined. One of the first studies that addressed this question evaluated the role of breast MRI in 167 patients diagnosed with DCIS in the final surgery. The preoperatively performed MRI showed 98% of the DCIS lesions by only missing two, which were diagnosed in the MG [16]. Another study about women in a screening program for women with a higher risk of breast cancer showed longer survival of women taking part in MRI screening than those without. They reported a sensitivity of MRI plus mammography of 93% and a specificity of 63% [18]. Saadatmand et al. showed that annual breast MRI and mammography combined could improve breast cancer metastasis-free survival for women with genetic or familial predisposition [17]. Diagnostic accuracy is comparable to our study, but the study cohort is significantly different from the patient collective studied here.

In our study, MG showed 59 suspicious MC findings in 58 patients (BI-RADS^®^ category four). Of these, 21 were malignant, corresponding to a PPV of 35%. This is comparable to the literature, where the PPV for suspicious MCs in MG is between 18.7% and 28.8% [24]. Around 64% of the histologically clarified suspicious MCs turn out to be benign [25,26]. Li et al. described a specificity of 42.9% and an accuracy of 69.2% for BI-RADS^®^ category 3–5 MCs [27]. Cilotti et al. describe a specificity of 59% and a PPV of 63% [6] for BI-RADS^®^ category four and five findings. In a study by Akita et al., MG showed a specificity of 24% and an accuracy of 44% [28] for all patients with suspicious MCs. In our study, 38 out of 59 MC findings were benign.

Surprisingly, we found five additional mammographically occult carcinomas in our cohort. This might be because of a close-to-the-chest wall localization outside the area recorded by the MG (2 cases). In the other three cases, women had extremely dense breasts, which is known to reduce the sensitivity of MG to under 30% [29]. Women with a breast density above 75% have a 4.7 times increased risk of breast cancer and carcinoma is detected 3.5 times worse in screening [30,31]. The five invasive carcinomas additionally detected by the breast MRI were hormone receptor positive and human epidermal growth factor receptor (HER) two negative. All patients were lymph-node-negative. Compared to the literature, the breast MRI detects especially early, node-negative, ductal and almost exclusively hormone-receptor-positive breast cancers [11,32]. In particular, two of the four patients with a high proliferation index (ki67) (20% and 60%) and lymph node negativity have benefited predictably from the earlier discovery by breast MRI [33]. In the literature, MRI screening of the contralateral breast in women with newly diagnosed breast cancer could improve overall survival with a hazard ratio of 2.51 [34]. One of the discussed limitations of breast MRI is overdiagnosis and therapy. We found nine additional lesions resulting in nine additional biopsies showing five invasive carcinomas. Nevertheless, four biopsies resulted in benign findings. However, in comparison, there is also a large overdiagnosis in MG with a PPV of BI-RADS category four microcalcifications described between 20 and 63% in the literature [4,6]. In the cohort analysis of Bennani-Baiti and coworkers, benign histology resulted in 60% of women who underwent a biopsy. In comparison, pooled PPV for BI-RADS^®^ category four microcalcifications in MRI is 81% [3].

On the other hand, for the MC evaluation, when performing an additional MRI, 32 biopsies could have been avoided, when both image modalities would have been considered. This corresponded to a possible avoidance of 54%, in absolute numbers 32 out of 59, of biopsies. Without biopsies, 3/21 (14%) of the DCIS would have been overlooked (two low-grade and one intermediate-grade DCIS). High-grade DCIS or invasive carcinomas have not been overlooked. In 2018, Baltzer et al. showed that 39.5% of 81 biopsies in patients with suspect MCs could have been avoided in 858 retrospectively evaluated patients. A total of 2/71 intermediate-grade DCIS were not found, representing 2.8%—compared to 3/21 (14%) in the present study [35]. Narod et al. showed a very small impact on long-term mortality of 3.3% in 20 years of DCIS alone [36]. Feinberg et al. and Sagara et al. also describe a frequent overdiagnosis and therapy of DCIS [37,38]. In addition, it has been shown that there is no survival benefit of surgical therapy for patients with low-grade DCIS [38].

One limitation of our study might be the size of the study, which also limits the power of statistical results as PPV. The size of the cohort was strongly regulated by the scanning time of the MRI but it corresponds to the study sizes used in the literature for similar study setups [3]. This could lead to a limitation of the significance of the statistics. Larger studies with a larger patient collective would be desirable.

## 5. Conclusions

In our study, an additional performed breast MRI could have increased the diagnostic reliability in the assessment of MC in screening MG. The PPV increased from 36% to 75% if both modalities had been considered for the final report. By performing an additional breast MRI for MC evaluation, 54% of the biopsies could have been avoided retrospectively in our study, by overlooking only two low-grade and one intermediate-grade DCIS. Further, the benefit of these MRIs in our small cohort was that a not inconsiderable number of malignant lesions without mammographically visible MCs was revealed.

## Figures and Tables

**Figure 1 diagnostics-13-01197-f001:**
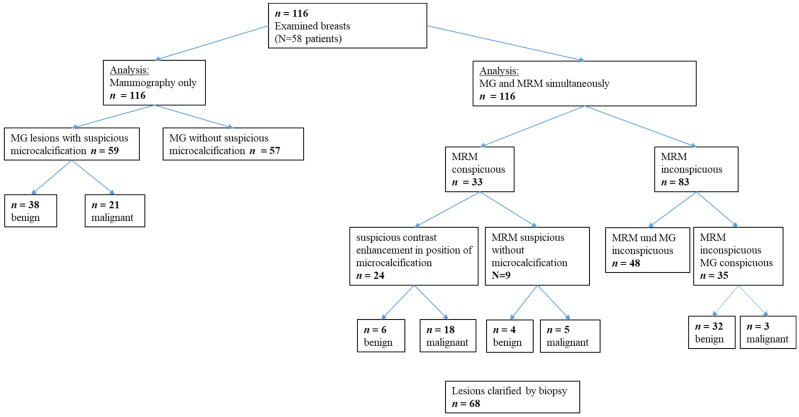
Results of the clinical evaluation as an overview. Breast MRI (magnetic resonance imaging), mammography (MG), *n* (number).

**Figure 2 diagnostics-13-01197-f002:**
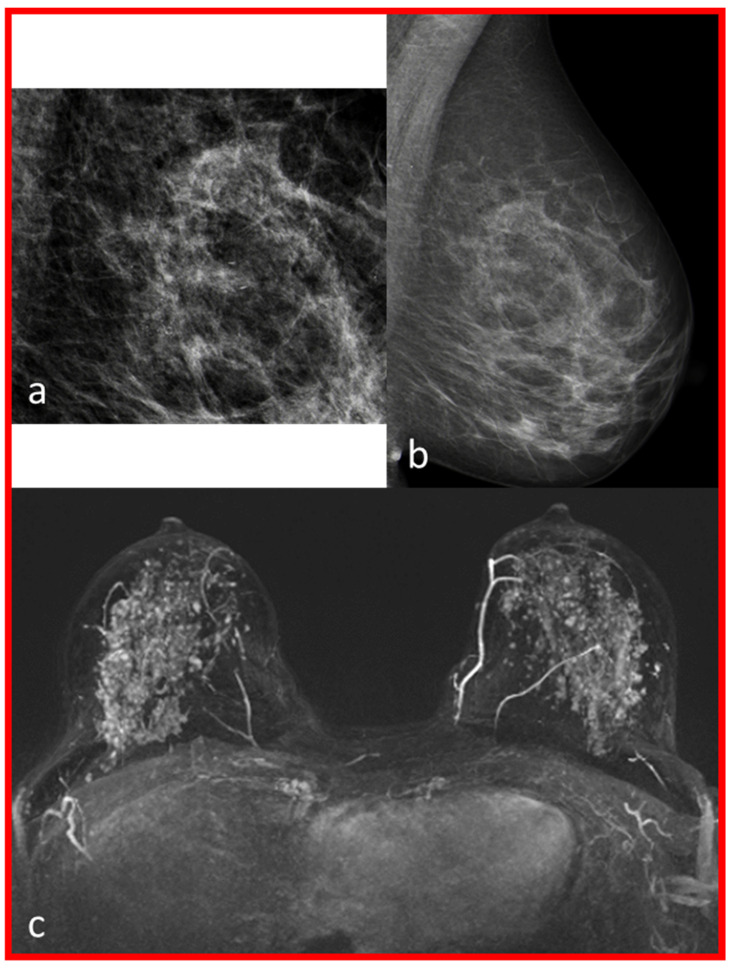
The mammography and breast magnetic resonance images (MRI) of patient number two of Table 3 are shown. Amorphous, regional MCs of the left breast on the left in MLO (mediolateral-oblique)—projection (images (**a**,**b**)). Early post-contrast phase MIP (maximum intensity projection): no suspicious lesion identifiable on both sides, possibly due to the strong BPE (background parenchymal enhancement) assessed as pronounced (image (**c**)).

**Figure 3 diagnostics-13-01197-f003:**
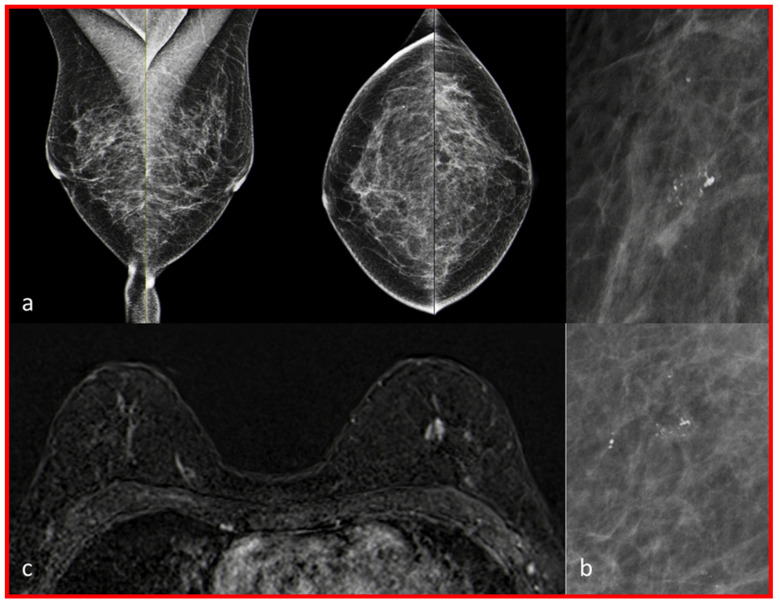
The mammography and breast magnetic resonance images (MRI) of patient number five of Table 2 are shown. Grouped, coarse, heterogeneous microcalcifications (MCs) in the mammography (MG) of the right breast (subfigure (**b**), enlargement of the microcalcifications of subfigure (**a**), histology: low-grade DCIS). Inconspicuous MG of the left breast. In breast magnetic resonance imaging, no suspicious lesion was detected in the right breast. A non-mass enhancement was found in the left breast (image (**c**), histology: breast cancer, NST G2).

**Table 1 diagnostics-13-01197-t001:** Characteristics of women with benign and malignant findings in mammography.

	Benign Lesions (*n* = 36)	Malignant Lesions (*n* = 22)	(*p*-Value)
Age in years Median (range)	57 (45–74)	57.5 (45–80)	0.290
Menopausal status Premenopausal *n* (%) Postmenopausal *n* (%)	6 (15.4%) 30 (84.6%)	6 (28.6%) 16 (71.4%)	0.306
Average BMI (Kg/m²) (SD)	25.72 (5.26)	25.98 (3.35)	0.855
Breast density *n* (%) a b c d	3 (8.33%) 11 (30.56%) 19 (52.78%) 3 (8.33%)	3 (13.64%) 12 (54.55%) 6 (27.27%) 1 (4.55%)	0.048 *
Average number of children (SD)	1.67 (0.92)	1.67 (1.07)	1.000

Characteristics of women with benign and malignant findings in mammography, kg (kilograms), m (meters), SD (standard deviation), *n* (number), BMI (body mass index), ACR (American College of Radiology), * (statistically significant).

**Table 2 diagnostics-13-01197-t002:** Additional findings detected by breast magnetic resonance imaging (MRI.).

Patient	MA	BPE	Breast MRI	D	MG	Modality	Histology/IHC/TNM
1	I	Mild	Mass, circumscribed, oval	c	Localization outside the MG coverage near the chest wall	Core biopsy guided by ultrasound	1.4 cm G1 NST/ER 90%, PR 30%, Ki-67 5%, HER2 negative/pT1c pN0 L0 V0 Pn1 G1
2	II	Moderate	NME, segmental, heterogeneous	b	No MCs	MRI-guided vacuum-assisted biopsy	13.4 cm low-grade DCIS
3	II	Moderate	NME, linear	b	Unilateral MCs in different position	MRI-guided vacuum-assisted biopsy	0.6 cm G1 tubular carcinoma/ER 90%, PR 30%, KI-67 10%, HER2 negative/pT1b pN0 L0 V0 Pn0 G1 associated with 3.2 cm low-grade DCIS
4	I	Minimal	Mass, irregular, not circumscribed-spiculated, heterogeneous	b	Unilateral MCs in different position	Core needle biopsy	0.5 cm G3 NST/ER 90%, PR negative, Ki-67 60%, HER2 negative/ypT0 pN0 L0 V0 Pn0
5	II	Moderate	NME, grouped, regional	c	No MCs	Partial excision	0.7 cm G2 NST//ER 90%, PR 40% Ki-67 20% HER2 negative/pT1b pN0 L0 V0 Pn0 G2 associated with 5.0 cm low-grade DCIS
6	I	Mild	NME, linear, heterogeneous	b	No MCs	Duct excision	Fibrocystic changes
7	I	Mild	NME, diffuse, heterogeneous	b	No MCs	Core needle biopsy guided by ultrasound	UDH
8	I	Minimal	NME, linear, homogeneous	c	Unilateral MCs in different position	Partial excision	Fibrocystic changes
9	I	Moderate	NME, focal, homogeneous	c	Unilateral MCs in different position	Partial excision	FEA

Additional findings detected by breast magnetic resonance imaging (MRI). MA (motion artifact), BPE (background parenchymal enhancement), NME (non-mass-enhancement), MG (mammography), ICH (immunohistochemistry), TNM (tumor/nodal/metastasis), ER (estrogen receptor), PR (progesterone receptor), KI-67 (proliferation marker), HER2 (human epidermal growth factor receptor 2), *p* (pathological), L (lymphatic), V (vessel), Pn (perineural sheath), UDH (usual ductal hyperplasia), FEA (flat epithelial atypia), DCIS (ductal carcinoma in situ), NST (breast cancer no special type), D: mammographic density.

**Table 3 diagnostics-13-01197-t003:** False negative findings in the combined analysis of mammography (MG) and breast magnetic resonance imaging (MRI).

Patient	MA	BPE	ACR	Histology and Size	Shape and Distribution
1	2	Moderate	B	Intermediate-grade DCIS (6.5 cm)	No contrast enhancement in the position of macrocalcifications, multiple small foci on both sides.
2	1	Marked	C	Low-grade DCIS (0.15 cm)	No contrast enhancement in the position of macrocalcifications.
3	1	Marked	C	Low-grade DCIS (5.6 cm)	No contrast enhancement in the position of macrocalcifications, multiple cysts on both sides.

False negative findings in the combined analysis of mammography and breast magnetic resonance imaging (MRI), motion artifact (MA), BPE (background parenchymal enhancement), ACR (American College of Radiology), DCIS (ductal carcinoma in situ) and breast MRI (magnetic resonance imaging).

## Data Availability

The data presented in this study are available upon request from the corresponding author.

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
