# Peer review of "Low-Risk Women with Suspicious Microcalcifications in Mammography—Can an Additional Breast MRI Reduce the Biopsy Rate?"

_diagnostics, 2023, doi:10.3390/diagnostics13061197_

Round 1

Reviewer 1 Report

I read with interest the manuscript titled "Low Risk Women with suspicious microcalcifications in mammography - Can an additional breast MRI reduce the biopsy rate?"; . While not entirely novel (you correctly cited a 2016 meta-analysis), it addresses a significant clinical problem. The novelty could be improved by specifically evaluating ultra-fast or abbreviated MRI which of course would be subject of another study.

1_In the patient selection section, the methodology explanation section could be improved by adding the total number of microcalcifications and BI-RADS 3 and 5 microcalcifications, which I believe were excluded from the study.

2_Also describing the shape and distribution of all three false negative MRI's would add some insight.

3_Figure 1 diagram is confusing and I do not totally understand the first arm of the algorithm.

4_In the statistics section, since I found no quantitative values compared, the use of two-sided independent t-test was not clear.

5_In the discussion section, it would be nice to acknowledge the few MRI overdiagnoses and compare the number of MRI overdiagnoses with the mammography overdiagnoses.

Reviewer 2 Report

The topic is very interesting. According to my opinion every suspicious lesion must be biopted. We prefer to use tomosynthesis instead of simple mammography. We obligatory use mammography in case of lobular carcinoma or is case of multifocal and multilocular tumors. 

58 patients' data seems very little amount to say the PPV.

References must be updated, they are really old.

Following corrections I am pleased to review again.

Author Response

Revision_2 – Repley to reviewer 2

Dear Editors,

Dear reviewer,

Dear ladies and gentlemen,

We are grateful for your comprehensive review and we appreciate your effort in time and your instructive and helpful comments to improve our manuscript. We have revised our manuscript according to your instructions. We hope you are satisfied with the changes of our manuscript. If further revisions should be requested, we will do so. Please find below our point-for-point replies to your comments.

Yours sincerely,

Patrik Pöschke on behalf of the authors

Reviewer 2

The topic is very interesting. According to my opinion every suspicious lesion must be biopted. We prefer to use tomosynthesis instead of simple mammography. We obligatory use mammography in case of lobular carcinoma or is case of multifocal and multilocular tumors. 

Reviewer’s comment

Authors’ reply

58 patients' data seems very little amount to say the PPV.

We fully comprehend this point of criticism and have added this issue to the limitations (page 11, lines 322-323).

References must be updated, they are really old

We update our references carefully, and added some recent publications.